

**Impact of climate change on the production and transport of**
**sea salt aerosol on European seas**
**J. Soares[1], M. Sofiev[1], C.Geels[2], J.H.Christensen[2], C.Andersson[3], S. Tsyro[4],**
**J.Langner[3]**
[1] Finnish Meteorological Institute, Helsinki, Finland
[2] Department of Environmental Science, Aarhus University, Roskilde, Denmark
[3] Swedish Meteorological and Hydrological Institute, Norrköping, Sweden
[4] EMEP MSC-W, Norwegian Meteorological Institute, Oslo, Norway
Correspondence to: J. Soares (joana.soares@fmil.fi)
**Abstract**
The impact of climate change on sea salt aerosol production, dispersion, and fate over the Europe is
studied using four offline regional chemistry transport models driven by the climate scenario SRES
A1B over two periods: 1990-2009 and 2040-2059. The study is focused mainly on European seas:
Baltic, Black, North and Mediterranean. The differences and similarities between predictions of the
individual models on the impact on sea salt emission, concentration and deposition due to changes
in wind gusts and seawater temperature are analysed. The results show that the major driver for the
sea-salt flux changes will be the seawater temperature, as wind speed is projected to stay nearly the
same. There are, however, substantial differences between the model predictions and their
sensitivity to changing seawater temperature, which demonstrates substantial lack of current
understanding of the sea-salt flux predictions. Although seawater salinity changes are not evaluated
in this study, sensitivity of sea-salt aerosol production to salinity is similarly analysed, showing
once more the differences between the different models. An assessment on the impact of SSA to the
radiative balance is presented.



## 1 Introduction

The sea salt aerosol (SSA) affects the Earth radiation budget, atmospheric chemistry, cloud processes, and climate (O'Dowd et al., 1997; IPCC, 2013). Anthropogenic and natural aerosols have similar annual impacts on the global radiative balance, though being predominant in different locations (Textor et al., 2006). SSA dominates the particulate mass and it is the major contributor to aerosol optical depth (AOD) over the ocean (Quinn et al, 1998).

SSA originates from sea spray droplets resulting from waves breaking on the seawater surface, forming whitecaps that cause the entrainment of air into the water. The two main mechanisms responsible for sea spray formation are air bubble bursting during whitecap formation and decay, and direct tearing of droplets from the top of breaking waves. Therefore, the formation of primary SSA is mainly dependent on wind speed: the emission of SSA is generally considered to be proportional to surface winds cubed (Monahan et al., 1986), suggesting that small changes in surface winds can have a substantial impact on the emission of this natural aerosol. Further on, studies on the marine aerosol size distribution (e.g. Covert et al., 1998; Russell and Heintzenberg, 2000; Bates et al., 2002; Huebert et al., 2003) suggest that for high wind speeds the production of very coarse SSA (with particle diameter ($D_p$) > 20 mm) increases, contributing to a higher transfer of heat and water vapour from the ocean to the atmosphere (Andreas et al., 1995). These processes have a strong impact on the climate forcing. Other parameters influencing the formation of primary SSA have been identified, e.g., seawater temperature and salinity, atmospheric stability, and wave height and steepness (O'Dowd and Smith, 1993; Gong et al., 1997; Gong, 2003; Mårtensson et al., 2003; Lewis and Schwartz, 2004; O'Dowd and de Leeuw, 2007; Witek et al., 2007a, 2007b; Ovadnevaite et al., 2014). Laboratory studies by Mårtensson et al. (2003) and in situ measurements by Nilsson et al. (2007) show that for nano-sized particles, the aerosol number emission decrease with increasing seawater temperature, and for particles with $D_p$ > 100 nm, the number SSA increase with increasing seawater temperature; reflecting different sea spray formation processes. Seawater salinity also affects the droplet formation, where formation of particles with $D_p$ < 0.2 μm are not affected by salinity, but for larger $D_p$'s, salinity impact is substantial: higher salinity contributes to higher production (Mårtensson et al., 2003). The SSA removal processes are scavenging by precipitation and dry deposition (including gravitational settling). SSA has an effect on secondary aerosols formed by gas-to-particulate conversion process such as condensation and nucleation (binary homogeneous and heterogeneous) (Twomey, 1997). SSA serves as a sink for condensable gases and smaller aerosol particles, and serves also as a medium for aqueous-phase reaction of



reactive gases, e.g. $H_2SO_4$. This can lead to nucleation suppression for other components of the
marine aerosol and consequently change their size distribution, creating a feedback on climate.
Furthermore, SSA formation results in a size spectrum ranging from 0.01 to 100 μm, which can
lead to cloud formation. With increasing concentrations of cloud condensation nuclei, the cloud
microphysical properties change, i.e., the available water vapour is re-distributed over more
particles, on average resulting in smaller particle sizes, which in turn changes both cloud albedo and
precipitation (Latham et al., 2008, Lenton and Vaughan, 2009; Boyd, 2008; Korhonen et al., 2010,
Wang et al., 2011). With dry diameter lower than 1 μm, SSA can easily be transported for long
distances in the atmosphere, serving as a cloud seed outside of heavily clouded regions. The cloud
drop number concentration can be spatially different, depending on the wind speed, atmospheric
transport and particle loss via dry and wet deposition (Korhonen et al., 2010).
Changes in atmospheric transport pathways, precipitation patterns, and sea ice cover influence
transport, removal and distribution of SSA. The main features of the regional and global SSA
distribution and the climate impact on SSA production due to these physical drivers have been
discussed in studies such as Liao et al. (2006), Pierce and Adams (2006), Manders et al. (2010),
Sofiev et al. (2011), Struthers et al. (2011), and Tsyro et al. (2011). The understanding of sea spray
emissions has increased substantially but process-based estimates of the total mass and size
distribution of emitted sea spray particles continue to have large uncertainties (de Leeuw et al.,
2011). Chemical transport models (CTM) and general circulation models (GCM) estimates of sea
salt burden may vary over 2 orders of magnitude (Textor et al., 2006) and climate models disagree
about the balance of effects, ranging from little (Mahowald et al., 2006a) to a considerable
sensitivity to climate change (Bellouin et al., 2011). The difference between the available
estimations might be due to the wind speed predicted by the climate models, with little
understanding of how wind speed may change over the ocean in a warmer climate (IPCC, 2013).
The main goals of the current study are to assess the sensitivity of the production, surface
concentrations and removal of SSA to climate change. A multi-model approach using four state-of-
the-art offline CTMs was taken to assess the uncertainty/robustness of model predictions over
Europe. The sensitivity of simulated emission, concentration, and deposition of SSA to changes in
climate was evaluated by comparing a past (1990-2009) and a future (2040-2049) period. This
study is a follow-up to the climates studies of Langner et al. (2012) focusing on surface ozone and
Simpson et al. (2014) focusing on nitrogen deposition.





## 2   Methods


This study uses the same modelling structure as in Langner et al. (2012) for ozone and in Simpson
et al (2014) for nitrogen. We focus on the comparison of SSA simulations from three offline
European-scale CTMs - EMEP MSC-W, MATCH and SILAM - and one offline hemispheric CTM,
DEHM. The models were run through a past (1990-2009) and a future (2040-2059) climate
scenarios and the results for the European seas (Baltic, North, Mediterranean, and Black Seas) were
compared. The climate meteorology data from a GCM were used in a regional climate model
(RCM) and the hemispheric model DEHM. The regional models where driven by the downscaled
meteorology from the RCM and the boundary conditions from DEHM. The horizontal grid for
DEHM is 150x150 km$^2$ and for the regional CTMs identical to the RCM (ca. 50×50 km$^2$).
Throughout the paper, the SSA mass refers to the total mass of dry particles. Since the observations
measure sodium (Na$^+$) concentrations rather than total SSA mass, it is assumed that Na$^+$ mass
fraction is ~30% (Seinfeld and Pandis, 2006). Particle sizes are also provided for dry conditions
and, unless otherwise stated, the dry diameter $D_p$ ranges up to 10 μm.

### 2.1   Climate meteorology


Results of the global ECHAM5/MPIOM GCM (Roeckner et al., 2006), driven by emissions from
the SRES A1B scenario (Nakicenovic, 2000), were downscaled over Europe with the Rossby
Centre Regional Climate model, version 3 (RCA3) (Samuelsson et al., 2011; Kjellstrom et al.,
2011). The global ECHAM5/MPIOM model is defined in spectral grid T63, which at mid-latitudes
corresponds to a horizontal resolution of ca. 140×210 km$^2$. The horizontal resolution of RCA3 was
0.44°×0.44° on a rotated latitude-longitude grid, and data were provided with 6-hourly resolution.
The climate, as downscaled by RCA3, reflects the broad features simulated by the parent GCM, but
from earlier studies with the current setup it is clear that the global ECHAM5/MPIOM model
projects a slightly warmer and wetter climate over Europe than the regional model RCA3 (Langner
et al., 2012; Simpson et al., 2014).
The wind speed is higher over the ocean and can be up to two times slower, in average, over the
inner seas (Fig. 1, first panel on the left). Wind patterns are different between the Seas, with some
areas over individual seas being more affected by wind gusts than others: e.g. in the Mediterranean,
the wind speed is higher over the Levantine Sea than over other areas. For the wind speed, RCA3
predicts an stronger increase at the Norwegian Sea, Black Sea, Gulf of Bothnia (Baltic Sea) and
Aegean Sea (Mediterranean Sea) and a stronger decrease between Italy and Tunisia and Libya





(Mediterranean Sea) in the future period  (Fig. 1, first panel on the right). Nevertheless, the absolute
change is no more than 0.4 m/s. Trend analysis considering only marine grid cells for each sea (Fig.
S1 in supplementary material) shows that there is no significant trend between past and future
periods.
Typically, the surface water temperature is higher at southern latitudes. For the same latitude, the
Black and Mediterranean Seas have, in general, higher temperature than the Atlantic Ocean and the
Baltic Sea (Fig. 1, second panel on the left). RCA3 predicts a general increase of the water surface
temperature between the past and the future periods (Fig. 1, second panel on the left). The most
substantial changes are for the northern part of the Atlantic Ocean and for the Baltic Sea (maximum
1.17 ºC). Trend analysis for the monthly mean temperature is significant for all the European inner-
seas (Fig. S2 in supplementary material). The temperature is rising for all the seas with the highest
rise over the Black Sea and the lowest over the North Sea.
The precipitation tends to be higher over the ocean and lower over the inner seas. The lowest
precipitation amount is seen over the Mediterranean Sea; on an annual level the difference from the
ocean can be up-to two orders of magnitude (Fig. 1, third panel on the left). The climate model
predicts that the precipitation will strongly decrease over the Mediterranean and increase over the
Baltic and North Seas, whereas over different parts of Atlantic Ocean the opposite trends can
coexist (Fig. 1, third panel on the right). Trend analysis shows that none of the trends is significant
(Fig. S3 in supplementary material).
**2.2   SSA boundary conditions**
Sea salt concentrations (as fine and coarse modes, see the description of DEHM below) provided by
the hemispheric DEHM model, were used as lateral and top boundaries for the regional models. The
boundary values taken from DEHM were updated every 6 h and interpolated from the DEHM grid
to the respective geometry of each regional CTM. The DEHM model was driven by the global
ECHAM5-r3 meteorology, without the RCA-3 downscaling.
**2.3   Chemical transport models**
The models used in this study have been introduced in the previous studies: Langner et al. (2012)
and Simpson et al. (2014). Below, we focus on their handling of the production and removal of
SSA. All the SSA source functions in the current study are based on white-cap-area based
parameterizations of Monahan et al. (1986), for formation of super-micron particles and follow



152 Mårtensson et al. (2003) for the sub-micron aerosols. The difference between the various source

153 functions is the dependence on temperature and salinity for the SSA generation (**Table 1**).

### 2.3.1 DEHM

155 In DEHM the production of SSA at the ocean surface is based on two parameterization schemes

156 describing the bubble-mediated sea spray production of smaller and larger aerosols. In each time

157 step the production is calculated for seven size bins and thereafter summed up to give an aggregated

158 production of fine (with dry diameters <1.3 µm) and coarse (with dry diameters ranging 1.3-6 µm)

159 aerosols. For the fraction with dry diameters less than 1.25 µm a source function based on

160 Mårtensson et al. (2003) is used, while for sizes larger than that the Monahan et al. (1986) source

161 function is applied. They both include an $U_{10}^{3.41}$ dependency on wind speed and the production of

162 the smaller aerosols is also a function of the sea surface temperature. An ambient relative humidity

163 of 80% is assumed in the calculations and the size of the produced SSA is assumed to depend on the

164 salinity at the actual location. Here a monthly climatology of current day salinity on a 0.25°x0.25°

165 grid (Boyer et al., 2005) is applied for both time periods in focus in the current paper. Within the

166 atmosphere, the fine and coarse fraction of SSA is treated separately in terms of transport and

167 removal. Wet deposition includes in-cloud and below-cloud scavenging, while dry deposition

168 velocities are based on typical resistance methods for various land surface types (see Simpson et al.,

169 2003; Emberson et al., 2000). The fine and coarse fractions in the DEHM model are in the current

170 paper assigned the dry diameters of 1 µm and 6 µm.

171 DEHM is continuously validated against available measurements from e.g. the EMEP network and

172 an evaluation of an earlier version of the sea salt routine in DEHM showed that the model gives

173 satisfactory results for sea salt over Europe (Brandt et al. 2012).

### 2.3.2 EMEP MSC-W

175 The standard Unified EMEP model runs include sea salt particles with ambient diameters up to

176 about 10 µm, which mainly originate from the bubble mediated sea spray (Tsyro et al, 2011). The

177 parameterisation scheme for calculating sea salt generation in the EMEP model makes use of two

178 source functions for bubble-mediated sea spray production. The first one is a source function for sea

179 spray droplets at 80% relative humidity from Monahan et al. (1986) and the second one is a source

180 function for sea salt particles from the work of Mårtensson et al. (2003), which is formulated for a

181 salinity of 33‰. In the EMEP model, the SSA fluxes can be calculated for particle dry Dp ranging

182 from 0.02 to 12 µm, whereas operationally and for this work SSA with Dp up to 6 µm are included.



Mårtensson et al. (2003) parameterisation is applied for smaller size bins, while Monahan et al.
(1986) parameterisation is used for the coarser ones. From the fluxes of sea spray, the sea salt mass
is calculated assuming sea salt density of 2200 kg/m3. The total production rates of fine and coarse
sea salt are calculated by integrating the size resolved fluxes (7 in the fine and 3 in the coarse
fractions) over respective size intervals. In the model, generated SSA is assumed to be
instantaneously mixed within the model lowest layer at each time step. The transport and removal
of sea salt is described individually for the fine and coarse fractions in the EMEP model. Dry
deposition parameterisation for aerosols is calculated using a mass-conservative equation from
Venkatram and Pleim (1999). The dry deposition due to gravitational settling is size-dependent and
diameters of 0.33 and 4.8 μm are assumed for the fine and coarse SSA. . Wet scavenging is treated
with simple scavenging ratios, accounting for in-cloud and sub-cloud processes. The scavenging
ratios are assigned to crudely reflect the solubility of different aerosol components, and the size
differentiated collection efficiencies are used in sub-cloud aerosol washout.
The present sea salt parameterisation was shown to give the best overall results as compared to a
number of other source functions within the EMEP model (Tsyro et al., 2011). The model SSA
calculations are extensively evaluated against long-term observations (Tsyro et al., 2011; EMEP
Reports http://www.emep.int).

### 2.3.3 MATCH

The treatment of SSA production in MATCH is based on the parameterization of Mårtensson et al.
(2003) for dry particle sizes of up to 0.4 μm aerodynamic radius, and on Monahan et al. (1986) for
larger particle sizes. The temperature correction following Sofiev et al. (2011) is applied to the
estimates from the Monahan scheme. The number of bins is flexible, but in this study four size bins
were used with Dp ranges 0.02–0.1 μm, 0.1–1 μm, 1–2.5 μm, 2.5–10 μm. The production of sea
salt droplets is calculated assuming an ambient relative humidity of 80% and a particle density of
1150 kg/m3 and is integrated over each size bin while dry removal rates are calculated using the
geometric mean size in each bin. Dry deposition over land is following Zhang et al. (2001) while a
separate parameterization accounting for bubble burst activity is used over sea (Pryor and
Barthelmie, 2000). Sea salt is assumed to 100% activated or scavenged by hydrometeors in-cloud
while below-cloud scavenging is handled following Dana and Hales (1976). The distribution of
salinity in sea water is taken from NOAA (2013). Further details and evaluation of MATCH sea salt
simulations using observed meteorology can be found in Foltescu et al. (2005) and Andersson et al.

214  (2014).





### 2.3.4 SILAM

The SSA production via bubble-mediated mechanism takes into account the effects of wind speed, salinity, and water temperature and covers sea salt particles with dry diameter from 20 nm to 10 µm. The observations from the Mårtensson et al. (2003) study for seawater surface temperature 298 K and sea water salinity 33 ‰ were used to extrapolate the scheme from Monahan et al. (1986) to particle sizes down to 20 nm. To calculate SSA production for other water temperatures and salinities, correction factors are applied which were derived based on the experimental data of Mårtensson et al. (2003). The full description of the parameterisation in the SILAM model can be found in Sofiev et al. (2011). The description of the temperature correction in Sofiev et al. (2011) was changed. Currently, the water temperature reference for the unified shape function is 20 ℃, instead of 25 ℃ as referred in Sofiev et al. (2011). The shape function has been updated accordingly and the new shape function ($dF_0$/d$D_p$) for particles with Dp ranging from 0.01 to 10 µm is described below:

$$\frac{dF_0}{dD_p} = (1 + 0.05 * D_p) * \frac{\exp\left(\frac{-0.11}{D_p}\right)}{0.4 + \exp\left(\frac{-0.2}{D_p}\right)} * \frac{6 * 10^5}{\left(1 * 10^{-4} * D_p^2 + D_p\right)^3} * 10^{1.19 * \exp\left(-\left(\frac{0.35 - \lg D_p}{0.8}\right)^2\right)} \tag{1}$$

For the current study the spume droplet formation based on Andreas (1998) was included, with spume being supressed for 10m wind speed lower than 6 m/s. The production of sea salt droplets is calculated assuming a dry particle density of 2200 kg/m$^3$. The size distribution is described by flexible bins. Production is integrated over each size bin while dry and wet removal rates are calculated using mass-weighted mean diameter in each bin. Depending on particle size, mechanisms of dry deposition vary from primarily turbulent diffusion driven removal of fine aerosols to primarily gravitational settling of coarse particles (Kouznetsov and Sofiev, 2012). Wet deposition distinguishes between sub- and in-cloud scavenging by both rain and snow (Sofiev et al., 2006; Horn et al., 1987; Smith and Clark, 1989; Jylhä, 1991). Gravitational settling, dry deposition and optical properties take into account the particle hygroscopic growth. For the simulations, five bins with the Dp ranges of 0.01–0.1 µm, 0.1–1.5 µm, 1.5–6 µm, 6–15 µm; and 15-30 µm were used. The distribution of salinity in sea water is taken from NOAA (2013).

SILAM model has been evaluated against a wide range of observations and models utilizing the above described parameterization (Sofiev et al., 2011; Tsyro et al., 2011).



## 2.4  Model evaluation

Sea water is the predominant source of Na$^+$ in the atmosphere, which can be used as its tracer in most regions of Europe. Evaluation of the model predictions was performed via comparison with observations available from the EMEP network (Co-operative Programme for monitoring and evaluation of the long-range transmission of air pollutants in Europe, http://www.emep.int, Tørseth et al. 2012) that perform regular measurements across Europe. The observations include Na$^+$ concentration in aerosol and ion analysis of precipitation including Na$^+$. Concentration measurements are sampled daily by a filter pack sampler (cut-off at $D_p$ = ~10 μm), at 2 m height; the concentration in precipitation is mainly sampled by a "wet-only" sampler and, in a few places, with bulk collectors. The wet deposition of Na$^+$ is obtained by multiplying the weighted mean concentration by the total amount of precipitation in a daily basis. For more details about the sampling the reader is referred to e.g., Hjellbrekke and Fjæra (2009). These sampling methods do not distinguish if the sodium is originated from natural (e.g. mineral dust) or anthropogenic sources. In some regions there might be certain amounts coming from combustion processes and industry, but overall the contribution of anthropogenic sources to the sodium budget is low (van Loon et al., 2005).

The measurement data were averaged to monthly level with the minimum completeness requirement of 75% temporal coverage per month and per year, between 1990 and 2009. The CTMs predictions for the measurement sites satisfying the temporal criterion were averaged on a monthly basis over the 20 years. Since the model computations were driven by climate model fields, no temporal collocation was done. Therefore, the primary parameter considered was the monthly Na$^+$ concentrations averaged over the past period. Modelled values were obtained from the model's lowest layer mid-point, which is defined somewhat differently for each model (**Table 1**). No near-surface concentration profiling was made, with the exception of EMEP where concentrations are corrected to 3 m height, largely due to unreliable stability estimates based on climate-model fields.

The model performance was evaluated by the following statistical measures: bias, spatial Pearson correlation coefficient (R), root mean square error (RMSE), bias and standard deviation (SD) ratio (SD$_{model}$/SD$_{observations}$). The evaluation included Na$^+$ concentration in aerosols at 29 measurement sites and ion analysis of Na$^+$ wet deposition at 133 measurement sites, which we consider sufficient for computing the basic statistical scores and plotting scatter plots. The location of the measurement sites are shown in Fig. S4 in the supplementary material.



## 2.5   Radiative transfer modelling

The radiative transfer modelling was completed offline with the libRadtran software package for radiative transfer calculations (Mayer and Kylling, 2005). This tool calculates radiances, irradiances and actinic fluxes for the given optical properties. The Earth radiative balance results from the difference between the incoming (direct and diffusive-downwards) and outgoing (diffusive upwards) radiation. The impact of SSA is assessed by the difference between an atmosphere with SSA and without SSA, for the past and future periods. The calculations were defined at the top of the atmosphere (TOA), with wavelength ranging from 0.2 to ~4 µm, in order to compute the integrated shortwave irradiance. All the runs considered wet and icy clouds, with the cloud cover taken from the climate model RCA3 and optical properties taken from MODIS observations (Pincus et al. 2011). Monthly-basis observations from AQUA and TERRA obtained from 2002 to 2014 were averaged in order to have climatological cloud optical fields. These fields were the same for both past and future period calculations. Earth albedo information is included in the calculations and is obtained from the NASA model, GLDAS Noah Land Surface Model L4 (Rodell et al., 2004), on a monthly basis for the period between 1990 and 2012. This dataset was averaged to obtain climatological surface albedo fields, remaining the same for both past and future periods. The calculations for an atmosphere with SSA included the AOD computed by SILAM: the AOD at 550 nm was computed for the full size-spectrum of the SSA described in Table 1. SILAM's optical thickness predictions are based on size distribution and spectral refractive index of SSA (Prank, 2008). The AOD data was monthly-averaged for every hour in a day, for the past and future periods. This allowed taking into consideration the length of the day, since solar zenith angle is computed for every hour. The description of the runs and assumptions are provided in Table 2. This setting was chosen in order to reflect an atmospheric state closer to reality, since there were no other aerosols available for this study. Keeping the atmospheric and cloud conditions constant between the past and the future, will allow pinpointing the impact of the SSA on the radiative balance.

## 3   Results

### 3.1   Comparison with observations

Figure 2 and Figure 3 show the performance of the CTMs estimating $Na^+$ surface concentrations and wet deposition, respectively, during the past period; Table 3 and Table 4 complete the statistical evaluation of the models for the surface concentrations and wet deposition, respectively. The



models showed similar performance with quite high correlation coefficients varying from 0.71 up to
0.85 for the concentrations but substantially lower for wet deposition (from 0.24 up to 0.41). The
difference between the model performances is quite small and varying for the different scores. The
highest correlation with the concentration observations was shown by DEHM (0.85), which also
demonstrated the highest RMSE and bias originating from a stronger overestimation over the
regions with observed low concentrations. EMEP showed the lowest RMSE and bias, as well as one
of the best correlation factors. SILAM tends to overestimate the lowest observed values (positive
bias) whereas MATCH has a stronger underestimation of the highest values (negative bias).
Comparing the winter (December, January and February) and the summer (June, July and August)
seasons, one can notice that the models perform better in summer, with higher correlation and lower
bias. The observed winter time levels are likely harder to be reproduced due to stronger winds and
faster changing weather, which might not be captured by the climatological runs.
Comparison of $Na^+$ wet deposition with measurements shows low correlation and substantial under-
prediction. This is particularly true for the high-deposition observations, which resulted in a strong
negative bias for all the models. The evaluation of modelled precipitation was presented in Simpson
et al. (2014), Table 4, and shows an overestimation of precipitation in the RCA3 model (reginal
CTMs) and underestimation in the precipitation used in DEHM. The overestimation leads to an
overestimation of the deposition of SSA close to the sources. Consequently, less SSA reaches the
shore and the measurement sites. The second major reason for discrepancy is that the observed wet
deposition does not cut-off the size of the particles, i.e. SSA coarser than 10 µm is accounted for,
including the SSA produced in the surf zone. This mostly explains the large negative bias of the
models, which reported $PM_{10}$ only, and, to some extent, the low correlation. This is demonstrated
when comparing SILAM scores taking into account the full size range available (Dp = [0.01-30]
µm): accounting for the coarser aerosols strongly reduced the bias, correlation strongly improved,
and RMSE became slightly smaller. In summer, the scores are slightly better than in winter, but the
absolute values and importance of this removal process is smaller in summer time.
In Simpson et al. (2014), it was shown that CTMs driven by RCM meteorology are likely to
perform worse than they would with data from numerical weather prediction models. Nevertheless,
the current comparison showed that CTMs can predict mean concentrations and depositions within
~30% uncertainty (for depositions, prediction of full size range is a pre-requisite), whereas the
spatial distribution patterns are reproduced with correlation higher than 0.7 also when driven by
climate model meteorology.





### 3.2 Current and future climate SSA emissions

The annual SSA emission in the reference period predicted by DEHM, MATCH and SILAM is
shown in Figure 4 (left panel). EMEP did not have this variable as an output. As expected, all
models predict the highest emissions over the Atlantic Ocean, with the Mediterranean Sea being the
second highest source. MATCH predicted, in average, 25% higher emissions over the
Mediterranean than SILAM. The emissions are mainly driven by the wind and typically expressed
by the white-cap produced by the surface-winds via the Monahan and O'Muircheartaigh (1980)
parameterisation. This empirical power-law is taken by all models participating in this study and
suggests emission (E) to be proportional to the 10m-wind speed ($U_{10}$) to the power of 3.41: $E \approx$
$U_{10}^{3.41}$, the so-called wind-forcing. Consequently, the SSA emissions (Fig. 4, left panel) clearly
correlate with the wind-forcing (Figure 5, left panel), in particular over the open ocean. However,
the use of the same functional dependence and input meteorology does not guarantee identical
emission, as it will be discussed further on. MATCH and SILAM seem more sensitive to the wind-
forcing over the Mediterranean than DEHM, possibly due to the horizontal resolution difference
between the hemispheric and regional CTMs (e.g. the Mediterranean is not properly resolved by the
global climate model, the driver for DEHM). Apart from the wind forcing, laboratory studies have
shown the relation between the emissions of SSA and seawater surface temperature and salinity:
SSA mass will be higher at sea areas with higher surface water temperatures and salinity
(Mårtensson et al, 2003). The temperature and salinity dependencies are included in the
parameterizations, therefore, the models predict for the same wind forcing, higher emissions for
higher water temperatures: the Mediterranean and Black Seas (Fig. 1 and Fig. 4, left panel). The
effect of salinity is best seen in the Baltic Sea (salinity ~ 9 ‰), which has comparable wind forcing
to some areas of the Mediterranean and the Atlantic (salinity ~33 ‰) but lower emission. SILAM
and MATCH show the highest difference between the inner-seas with at least 3 times lower
emissions over the Baltic Sea.
In absolute terms, the climate impact on SSA emissions (Fig. 4, right panel) is mainly positive
according to the regional models whereas DEHM shows a general decrease. The exception goes for
the Atlantic Ocean, in the west side of the domain, where all the models agree in a decrease of
emissions. The difference between the past and future periods is only due to the wind forcing and
temperature changes, since salinity was kept constant. Thus, this change (Fig. 4, right panel) highly
correlates with the changes for wind-forcing (Fig. 5, right panel), adjusted by the changes in water
temperature (Fig. 1, right panel). For example, the pronounced decrease of emission over western



Atlantic is mainly driven by the reduction of wind speed but the decrease is limited by the rising
temperature in the north and east: higher temperature leads to production of more SSA even for
somewhat slower wind speed.
The models demonstrated different sensitivity to seawater temperature: it seems to be less important
for DEHM than for other models, whereas SILAM is the most sensitive. For instance, MATCH and
SILAM showed an increase of emissions over the east of Iceland where temperature is predicted to
rise by almost 2 K. The increase of seawater temperature, supported by higher wind speed, over the
Black and Aegean Seas (Fig. 1, right panel), will lead to higher emissions. DEHM might not be so
sensitive to the local storms due to the coarse horizontal resolution. The absolute difference
between future and past is the smallest for the Baltic Sea, but in relative terms all the models show
an increase up to 20% in Gulf of Bothnia, which is actually higher than, e.g. 5-15% of increase
predicted for North Sea (minimum for DEHM and maximum for MATCH).
Trend analysis for the Baltic, Black, Mediterranean and North Seas (only sea cells are taken into
consideration) is available as supplementary material: Fig. S5 for the Baltic, Fig. S6 for the Black,
Fig. S7 for the Mediterranean, and Fig. S8 for the North Seas. The trend is only statistically
significant ($p < 0.001$) for all the models for the Black Sea, with all models agreeing on an increase
of concentration in the future.
Figure 6 (left panel) shows the SSA emission difference between the winter and summer for the
past period. The difference between seasons in terms of SSA production can be substantial: SSA
emission is up to 3 times higher in winter time. Seasonally, there are differences between the
driving processes for SSA production: the winter period has a larger SSA production, due to more
frequent and stronger storms; but the summer time shows pronounced maxima over specific areas
mostly influenced by the seawater temperature. The latter is mostly true for MATCH and SILAM,
since their temperature sensitivity is higher. SSA emission in winter will be accentuated in the
future for MATCH (more emphasized) and SILAM: Figure 6 (right panel) shows pronounced
maxima around Iceland and the British Isles; distinct differences in the SSA emission are also seen
in the Mediterranean. DEHM does not show much difference between the periods.
**3.3   Current and future climate SSA concentrations**
Concentration is a function of emission and transport of the SSA, that is dependent on ventilation of
an area over inner seas (wind speed), and on removal processes largely controlled by precipitation
and relative humidity (via settling). Generally, the pattern of SSA concentration follows the





emission areas with stronger winds and frequent storms. Concentrations are, therefore, higher at the
Atlantic Ocean and lower at the European inner-seas. All the models show lower concentrations for
the Baltic Sea, reaching up to 10 times difference from the ocean (Fig. 7, left panel). The
Mediterranean Sea is the inner sea with the highest concentrations. For the Baltic Sea, DEHM and
MATCH show the highest and the lowest concentrations, respectively, with a difference of a factor
of ~1.3 between each other. For the Black Sea, DEHM and EMEP show the highest concentrations
and a similar spatial distribution pattern, and SILAM the lowest; nonetheless the difference is not so
substantial. For the Mediterranean Sea, EMEP shows the lowest concentrations, MATCH being the
highest: with 30% difference. All models show pronounced maximums at the Balearic Sea and the
Levantine Sea. Transport over land is quite similar among the models, especially for the regional
CTMs. The biggest difference lies over the western-central Europe with MATCH showing lower
concentration over land. Transport of SSA inland is visible hundreds of km's inland; near the coast
line it can contribute up to 6 $\mu g/m^3$ to $PM_{10}$.
The models predict relatively similar pattern for the SSA spatial distribution for the past period but,
they seem to have different responses to the future climate, with MATCH and SILAM clearly being
the most sensitive and EMEP the least. Figure 7 (right panel) shows the difference between the past
and future periods for the different models. DEHM and EMEP foresee almost no change or a
decrease of SSA concentrations over the open sea, whereas MATCH and SILAM predict an
increase. These results were expected due to the predicted emissions (Sect. 3.2). All models agree in
an increase in SSA surface concentration over the north of Iceland, the Black Sea, and over land in
southern latitudes. The models agree somehow on an increase of the Mediterranean and Black Seas
SSA concentration but it is MATCH and SILAM that show the highest positive change in
concentrations. The impact over land is slightly positive for all the models in the Southern part of
the domain, while at more Northern latitudes DEHM and EMEP from one side, and MATCH and
SILAM models from another, disagree on the trend signal: a reduction of the SSA load over land is
predicted by the first two models and an increase by the latter pair.
Overall, EMEP is the least sensitive and MATCH the most sensitive model to a changing climate.
SILAM is the most sensitive over the Norwegian Sea. The difference between the past and future
period concentrations is more substantial than that of emissions: the factors seemingly having
exacerbated this difference are the decrease of ventilation over the west-Mediterranean, changes in
mixing patterns, etc.



Trend analysis (supplementary material: Fig. S9 for the Baltic, Fig. S10 for the Black, Fig. S11 for
the Mediterranean, and Fig. S12 for the North Seas) suggest that trends are only significant ($p <$
$0.001$) for MATCH and SILAM for both Mediterranean and Black seas, all with a positive signal.
Seasonally, the concentrations follow the same pattern as the emissions: higher in winter time.
When analysing the changes between winter and summer, the models can again be grouped into
DEHM-EMEP and MATCH-SILAM. In winter (Fig. 8, left panel), the first pair presents a larger
amount of SSA mass generally over sea and land surfaces. Conversely, MATCH and SILAM
predict a decrease of SSA surface concentration around the British Isles, Mediterranean and Black
Seas, though the coast lines have sharper peaks of SSA mass during winter. The difference between
the future and past periods (Fig. 8, right panel) is relatively similar for all the models over the open
sea: predictions show an increase of concentration around the British Isles and a decrease over the
Norwegian Sea, in the future. MATCH and SILAM show sharper increase or decrease along the
Mediterranean Sea. The changes predicted can be 3 times higher than the changes predicted for the
emissions (Figure 6, right panel). The changes can also have different signal, e.g. the Eastern-basin
of the Mediterranean where it is predicted an increase of emissions but a decrease of concentrations,
implicating that the ventilation over this area was quite effective.
**3.4   Current and future climate SSA deposition**
The deposition (wet+dry) patterns for SSA are depicted in Fig. 9 (left panel). Typically the
deposition is higher over the sources areas and close to the coastal areas. Over land, SILAM shows
less deposition and DEHM and EMEP predict the highest levels. There are different patterns over
the Atlantic, mostly attributable to the boundary conditions treatment by each model. DEHM
predicts quite high values over all the seas. Over the Black Sea, the deposition is more accentuated
in the predictions by EMEP and less by SILAM. MATCH also shows higher values for deposition
over the Mediterranean, and SILAM the lowest. Deposition is not substantial over the Baltic Sea,
with exception of DEHM, owing to low SSA mass released from its surface.
The impact of future climate conditions (Fig. 9, right panel) on deposition, in absolute levels, is
small and mostly noticeable over the Atlantic Ocean. For all models, the most significant positive
change in the deposition is seen around Iceland. This is expected according to the changes seen in
precipitation between future and past periods (Fig. 1, third panel on the right). All regional CTMs
show a strong signal on the west side of the domain, an artefact due to the boundary conditions. In





relative terms, Scandinavia, east of UK, central-western Europe and Mediterranean are the most
affected with 5-20% more deposition predicted by MATCH and SILAM.
Trend analysis (supplementary material: Fig. S13 for the Baltic, Fig. S14 for the Black, Fig. S15 for
the Mediterranean, and Fig. S16 for the North Seas) suggests that none of models show a significant
trend.
Seasonally, SSA deposition is higher in winter than in summer, due to the higher emissions and
frequent precipitation in winter months. This difference is mainly accentuated over the source areas:
MATCH and SILAM have the lowest difference over the Baltic and Black Seas, due to the lower
production; DEHM shows the highest at Mediterranean Sea. The difference of deposition between
winter and summer will also change in the future period (Fig. 10, right panel) with all models
showing a slight increase of the deposition in summer over the Mediterranean and along the coast of
Norway. An increase of deposition in winter was suggested around Iceland and British Isles, North
Sea and coastal areas of Mediterranean Sea.

## 475    4    Impact of meteorology and seawater properties on the emission and fate of SSA

The multi-model comparison presented in Sect. 3 shows that there are significant difference
between the models in terms of emission and fate of the SSA. The latter is particularly true for the
inner seas. The differences between the models lead to a more uncertain answer about the impact of
the future climate on the production and transport of SSA and its possible feedback to climate. The
SSA emission in the models is driven by three parameters: wind speed, water temperature, and
water salinity. All models use the same $U_{10}^{3.41}$ dependence on wind speed; hence the differences in
emission have to be attributed to parameterization of temperature and salinity dependencies.
Formally, all models used the Monahan et al. (1986) and Mårtensson et al. (2003) parameterizations
or, at least, the available data for deriving the emission flux parametrizations (SILAM). Specifics of
the implementation, however, appeared significant. To understand the latter, box-model calculations
of the SSA mass flux as a function of temperature were made for seawater salinity 10 and 35 ‰,
representing Baltic Sea and Atlantic Ocean, respectively, and with wind-speed fixed at 15 m/s (Fig.
11, left-hand panel).
In general, all the models show an increase of mass flux of SSA with temperature and salinity,
except EMEP that does not apply any correction for salinity. Both DEHM and EMEP mass flux
show little difference between low and high temperatures; SILAM and MATCH show a substantial



dependency of the mass flux on temperature throughout the size ranges. This difference is explained
by the way dependency on seawater temperature is implemented: only for the fine mode in DEHM
and EMEP, based on the Mårtensson et al. (2003) source function, and for both fine and coarse
modes in SILAM and MATCH. In MATCH, the implementation of seawater temperature correction
is done by combining the temperature correction included in the Mårtensson et al. (2003) for size-
range below Dp = 0.4 μm and the use of the temperature corrections from Sofiev et al. (2011) for
the coarser sizes. In SILAM the source function is scaled with Sofiev et al. (2011) size-dependent
temperature correction function. This explains why the results in Sect. 3 could be paired between
the models. EMEP is the model that shows the highest amount of SSA produced, with the exception
for seawater temperature higher than 15 ºC and high salinity, with MATCH and SILAM predicting
the highest amount of SSA. For the lowest salinity, SILAM is the model that produces less SSA,
with DEHM being surpassed by MATCH around 17 ºC. For the highest salinity, both MATCH and
SILAM start to predict higher SSA flux than DEHM around 9 ºC. This is due to the temperature
correction factor described in Sofiev et al. (2011) that assumes that for low seawater temperature,
the production of coarse SSA, where the mass is significant, is very low. This analysis clarifies why
MATCH and SILAM tend to have higher emissions than DEHM where waters are warmer and
lower when colder (e.g. Baltic Sea), and why MATCH shows the highest values for the SSA mass
flux. Also explains the smaller difference between winter and summer predicted by DEHM, since
the changes in SSA mass flux depending on seawater temperature is very low.
Figure 11 (right panel) shows how the different models distribute the mass between the fine and
coarse modes, for the same wind and salinity conditions described above. Both DEHM and EMEP
assume that the contribution of the coarser mode is reduced with temperature, since more SSA is
produced with higher temperatures, for size ranges below 2.5 μm. EMEP has the highest
contribution for the coarse mode, independent of the temperature. For MATCH and SILAM, the
contribution to the coarser mode increases with temperature, though MATCH has a lower coarse
mode contribution than SILAM. The only agreement between the DEHM, MATCH and SILAM is
that for higher salinities, the coarse mode contribution is higher. The ratio between fine and coarse
mode is very relevant for the deposition processes, and it could explain why deposition is higher for
DEHM and EMEP (Fig. 9), though in this case, it is hard to evaluate the real impact due to different
deposition schemes implemented in the models.
It is pertinent to discuss the difference between DEHM, EMEP and MATCH, since these models
apply the same parameterization for SSA number flux, though having different salinity fields and



salinity correction function. Mårtensson et al. (2003) defines very strict size ranges for the
computation of the $6^{th}$ order polynomial for particles between 0.02 to 2.8 µm in dry diameter. In
case the models define size ranges outside of the tabulated in that study, it can result in very
different results. The linkage between the two parameterizations can also result in different
outcomes: DEHM links the two parameterizations at dry diameter of 1.25 µm, EMEP at 1.5 µm and
MATCH at 0.4 µm. In the case of MATCH, an extrapolation of the Monahan et al. (1996) function
is needed, in order to bring it to Mårtensson et al. (2003) range.

**5   SSA and climate change: production, fate and radiative impact**
The regional-scale impact of SSA production and fate caused by a changing climate has been
shown in Sect. 3. We show that the change in SSA emission between the past and future periods is
not so large, arguably due to the small change in wind speed between the two time periods.
Climates studies such as Gregow et al. (2011) projected higher wind speed changes in periods
closer to the years 2100, in Scandinavia. Nevertheless, the available climate estimations of wind can
differ substantially given the little understanding of how wind speed may change over the ocean in a
warmer climate (IPCC, 2013). Studies such as Salisbury et al. (2013) suggest that other variables, in
addition to wind forcing, influence the whitecap fraction, such as the seawater temperature or the
sea state. New parameterization for whitecap fraction, based on satellite observations, claims that
the whitecap-area based parameterization used by all the models in this study is misrepresenting the
absolute values. Albert et al. (2015) suggests that for higher latitudes the values are overestimated,
and underestimated for lower latitudes. If following that parameterization, the emission over the
Mediterranean is underestimated. This could mean that the changes in seawater temperature would
impact the SSA emission flux more substantially than suggested by this study.
The aerosol direct radiative effect (DRE) is defined as the difference between net radiative fluxes at
TOA in the presence and absence of SSA. The radiative forcing depends on the AOD of the aerosol
species in the atmosphere, the surface albedo and the vertical position of clouds. In this study, all-
sky conditions were considered, i.e. clouds are included. Over the seawater surfaces, SSA directly
scatters solar radiation back to space, resulting in a cooling effect on the climate by decreasing the
amount of radiation absorbed by the water surface. Over land, there can be both cooling over the
low-reflectance surfaces, and warming over high-albedo surfaces (e.g., Haywood and Boucher,
2000). Adding only a low absorbing aerosol, such as SSA, and assuming the same atmospheric and



cloud conditions for the all the runs (with and without SSA), the upward scattering by SSA will be
the only radiation impact in this study.
Figure 12 shows the DRE due to SSA in the past (left panel) and the change in DRE due to the
changing climate (right panel). These calculations are based on the AOD predicted by SILAM for
the past and future. As expected, the past computations predict the highest cooling effect due to
SSA over the areas where concentrations (Fig. 7, left-lower panel) are the highest and where the
surface albedo is the lowest (seawater surfaces). The strongest effect is seen over the Mediterranean
Sea due to the lowest cloud cover and the largest number of hours of sunlight per year. Studies such
as Ma et al. (2008) and Lundgren et al. (2013), state that the impact of clouds can be substantial,
reducing the direct radiative impact of SSA. The lowest cooling effect is predicted over land where
the albedo is higher and SSA amount is the lowest. Conversely, warming is predicted where the
albedo is high and the AOD is low, e.g. over the mountain tops in Norway and Italy. The current
study estimates the upward scattering by SSA, at TOA, to be up to 0.5 W m$^{-2}$ over seawater
surfaces. This value is within the estimates on upward scattering of radiation by SSA: ranging
between 0.08 and 6 W m$^{-2}$, at wavelengths in the range of 0.3-4 μm (Lewis and Schwartz, 2004).
Figure2, right panel, depicts the change in the DRE due to SSA between future and past. The results
suggest overall cooling (negative change) in the future: North of Iceland, Norwegian and North
Seas are the areas where the cooling is more accentuated. The Mediterranean area seems to be again
the most sensitive area in our study: it is predicted an overall warming for this area, both over sea
and over land, but also cooling, in particular in the east of the eastern basin. DRE pattern for the
whole year is highly influenced by the summer period due to largest number of daylight hours. This
can be seen in Fig. 13, right panel, which shows the change between future and past but considering
only the summer months (JJA). This study predicts a substantial seasonal variation for the DRE in
the sea surface waters. This is expected due to the variation shown in Sect. 3.2 and 3.3. The upward
scattering in the summer time can be up to 1.7 times higher than in winter, due to lower cloudiness
and lengthier daylight.
Figure 13 shows the change in winter (left panel) and summer (right) between the future and the
past. The strongest impact in winter is seen over the Mediterranean area: negative over the sea
surface and positive over land. In summer, the highest impact is over the seawater surfaces,
predicting a cooling effect in the future, with exception over the western basin of the Mediterranean
and the western side of the British Isles and France.





The results presented in this study for the present period are in accordance with the regional
simulations for a summer month presented by Lundgren et al (2013) and the global simulations
presented by, e.g. Grini et al. (2002) and Ma et al. (2008). The results are shown in Table 5.
The radiative forcing calculation is also sensitive to the SSA single scattering albedo. Thus, setting
the SSA's single scattering albedo as low as 0.95 (Russel et al, 2002), leads to a wide areas over
land where warming is substantial: essentially, over all surfaces with albedos exceeding 0.5 and low
(<0.03) aerosol load (not shown). We have chosen to show results for a more realistic SSA single
scattering albedo of 0.99 (Lundgren et al., 2013).

**6    Conclusion**
This study has compared predictions of SSA emissions, surface concentration and deposition from
four CTMs for both current condition and future scenarios, focusing on the European Seas: Baltic,
North, Mediterranean, and Black Seas. The three European-scale CTMs (EMEP, MATCH and
SILAM) were driven by the regional climate model (RCA3) meteorology and by the hemispheric
model (DEHM) boundary conditions. The hemispheric model was driven by the ECHMA5
meteorology. The impact of climate change on SSA production and fate, due to changes in wind
speed and seawater temperature, was analysed. Additionally, consideration about the impact of
seawater salinity on emissions was given.
The impact of climate change on SSA production and fate has different response from the models,
with the similar results between DEHM and EMEP, and between MATCH and SILAM. DEHM-
EMEP show almost no difference between future and past periods, and MATCH-SILAM shows a
general increase of the emissions and surface concentrations with levels reaching 30% in change.
The emissions increase is substantial in the Black Sea, Gulf of Bothnia (Baltic) and Levantine Sea
(Mediterranean), correlating well with the wind-forcing ($\approx U_{10}^{3.41}$) computed with the changes
predicted between the same periods. Nevertheless, the major driver of the changes of the sea-salt
fluxes from the sea surface will be the changing seawater temperature, since near-surface wind
speed is projected to stay nearly the same in the climate scenario used, in absolute levels the wind
will change less than a meter per second, in average, between the two periods. The concentrations
are predominantly increasing in Black and Mediterranean Sea. The impact of climate change on
SSA on deposition is not really relevant; though an increase is projected around Iceland by all the
models. Boundary conditions impact on the predictions is substantial.





The discrepancies between the models raised additional question about the implementation of the
SSA production formulations, since three of the models are based on the same parameterizations.
This study shows that the way a given parameterizations is implemented in the models and the
temperature and salinity correction functions play an important role for the final scaling of the SSA
flux: size range prescription may play a substantial role on the SSA flux calculation.
Simple calculations with the libRadTran allowed understanding the impact of SSA on the direct
radiative forcing. According to this study the upward scattering by SSA, at TOA, can to be up to 0.5
W m$^{-2}$ over the seawater surfaces in the present period, predicting an overall cooling in the future.
The most affected areas by cooling will be North of Iceland, Norwegian and North Seas, and the
eastern basin of the Mediterranean; warming is predicted manly in Mediterranean Sea, including
over land.

### Acknowledgements

This study was supported by the Nordic Council of Ministers, EnsCLIM and CarboNord projects.
The authors also thank Antti Arola for his guidance in the radiative forcing calculations and
interpretation of the results.

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



Table 1 Model characteristics for SSA computations.

| model | mode | Dp [μm] | source function | dependency | humidity | Lowest model layer thickness (m) |
|---|---|---|---|---|---|---|
| **DEHM** | fine | <1.3 | MA02 | T S | static (80%) | 60 |
| | coarse | [1.3-10] | MO86 | S | | |
| **EMEP** | fine | <2.5 | MA02 | T | static (80%) | 90 |
| | coarse | [2.5-10] | MO86 | - | | |
| **MATCH** | fine | [0.02–0.1] | MA02 | T S | dynamic | 60 |
| | | [0.1–1] | | | | |
| | | [1–2.5] | | | | |
| | coarse | [2.5–10] | MO86 | T (SO11) S | | |
| **SILAM** | fine | [0.01–0.1] | SO11 | T S | dynamic | 25 |
| | | [0.1–1.5] | | | | |
| | coarse | [1.5–6] | | | | |
| | | **[6–15]** | SO11 | T S | | |
| | | **[15-30]** | | | | |

**T**: temperature, **S**: salinity, **MO86**: Monahan et al. (1986); **MA03**: Mårtensson et al. (2003), **SO11**:
Sofiev et al. (2011). In bold, the modes not used for the PM$_{10}$ analysis.





Table 2 Assumption for the radiative transfer modelling libRadTran2.0 for present and future.

| | | |
|---|---|---|
| **Clouds** **(icy and wet)** | cloud cover | monthly averaged RCA3 fields (1990-2009); same for both periods |
| | AOD | monthly averaged MODIS data (2002-2014) (Pincus et al. 2011); same for both periods |
| | vertical profiles | wc.dat*; wc.dat* |
| **Atmospheric** **properties** | vertical profiles | subarctic winter, latitude over 60°: afglsw.dat* |
| | | subarctic summer, latitude over 60°: afglss.dat* |
| | | mid-latitude winter, latitude below 60°: afglmw.dat* |
| | | mid-latitude summer, latitude below 60°: afglms.dat* |
| | altitude, pressure and temperature | monthly averaged RCA3 fields (1990-2009); same for both periods |
| **Aerosol** **properties** | vertical profile | aerosol_default* |
| | AOD | dynamic: SILAM AOD calculations |
| | asymmetry factor | 0.8 (Ma et al. 2008) |
| | single scattering albedo | 0.99 (Lundgren et al, 2013) |
| | angstrom coefficient | 0.2 (Kaskaoutis et al, 2007; Kusmierczyk-Michulec & van Eijk, 2009) |
| **solar zenith angle** | | dynamic: computed with libRadTran sza tool |
| **surface albedo** | | monthly averaged NOAA data (1990-2012) (Rodell et al., 2004); same for both periods |
| **RTE solver** | | DISORT |
| **integrated shortwave calculation scheme** | | KATO2 (wavelength ~[0.2, 4] μm) |

*standard file in libRadTran



Table 3 Statistical evaluation of model results for surface SSA concentration (Na$^+$µg m$^{-3}$),
considering the whole year (annual), winter (December, January and February) and summer
periods (June, July and August), for 33 EMEP measuring sites, between 1990 and 2009.

| | annual | winter | summer | annual | winter | summer |
|---|---|---|---|---|---|---|
| **Obs** | 0.72 | 0.94 | 0.55 | | | |
| **DEHM** | 1.08 | 1.39 | 0.74 | | | |
| **EMEP** | 0.64 | 0.75 | 0.49 | | | |
| **MATCH** | 0.45 | 0.42 | 0.42 | | | |
| **SILAM** | 0.86 | 0.78 | 0.94 | | | |
| | | correlation | | | StdRatio | |
| **DEHM** | 0.85 | 0.87 | 0.81 | 1.72 | 1.57 | 1.79 |
| **EMEP** | 0.82 | 0.84 | 0.80 | 0.69 | 0.54 | 0.85 |
| **MATCH** | 0.75 | 0.82 | 0.77 | 0.48 | 0.33 | 0.66 |
| **SILAM** | 0.71 | 0.77 | 0.75 | 1.05 | 0.75 | 1.59 |
| | | RMSE | | | Bias | |
| **DEHM** | 0.97 | 1.11 | 0.70 | 0.36 | 0.45 | 0.18 |
| **EMEP** | 0.53 | 0.75 | 0.36 | -0.08 | -0.18 | -0.06 |
| **MATCH** | 0.69 | 1.03 | 0.41 | -0.27 | -0.52 | -0.14 |
| **SILAM** | 0.71 | 0.76 | 0.74 | 0.14 | -0.16 | 0.38 |







Table 4 Statistical evaluation of model results for SSA wet deposition ($Na^+mg\ m^{-2}$), considering
the whole year (annual), winter (December, January and February) and summer periods (June, July
and August) for 133 EMEP measurement sites, between 1990 and 2009. SILAM5m is the
evaluation if considering the whole possible size range for SSA Dp = [0.01-30] µm.

|  | annual | winter | summer | annual | winter | summer |
|---|---|---|---|---|---|---|
| **obs** | 1.59E+06 | 6.88E+05 | 1.36E+05 | | | |
| **DEHM** | 1.41E+06 | 5.59E+05 | 1.40E+05 | | | |
| **EMEP** | 1.64E+06 | 6.44E+05 | 1.65E+05 | | | |
| **MATCH** | 6.08E+05 | 1.77E+05 | 9.64E+04 | | | |
| **SILAM** | 8.42E+05 | 2.81E+05 | 1.25E+05 | | | |
| **SILAM5m** | 1.70E+06 | 6.70E+05 | 1.83E+05 | | | |
|  | | correlation | | | StdRatio | |
| **DEHM** | 0.55 | 0.53 | 0.41 | 0.36 | 0.31 | 0.55 |
| **EMEP** | 0.38 | 0.32 | 0.33 | 0.47 | 0.44 | 0.53 |
| **MATCH** | 0.49 | 0.50 | 0.34 | 0.13 | 0.11 | 0.26 |
| **SILAM** | 0.49 | 0.45 | 0.38 | 0.22 | 0.19 | 0.41 |
| **SILAM5m** | 0.62 | 0.63 | 0.37 | 0.86 | 0.84 | 0.93 |
|  | | RMSE | | | Bias | |
| **DEHM** | 3477 | 5513 | 866 | -114 | -327 | 10 |
| **EMEP** | 3778 | 6006 | 912 | 34 | -112 | 74 |
| **MATCH** | 3879 | 6122 | 892 | -634 | -1304 | -102 |
| **SILAM** | 3737 | 5945 | 871 | -483 | -1038 | -29 |
| **SILAM5m** | 3335 | 5070 | 1032 | 73 | -44 | 122 |







Table 5 Predicted direct radiative effect (W m$^{-2}$) by SSA for the past period

|  | annual | winter | summer |
|---|---|---|---|
| **sea** | -0.25±0.22 | -0.077±0.053 | -0.21±0.012 |
| **land** | -0.20±0.18 | -0.073±0.0019 | -0.083±0.0030 |






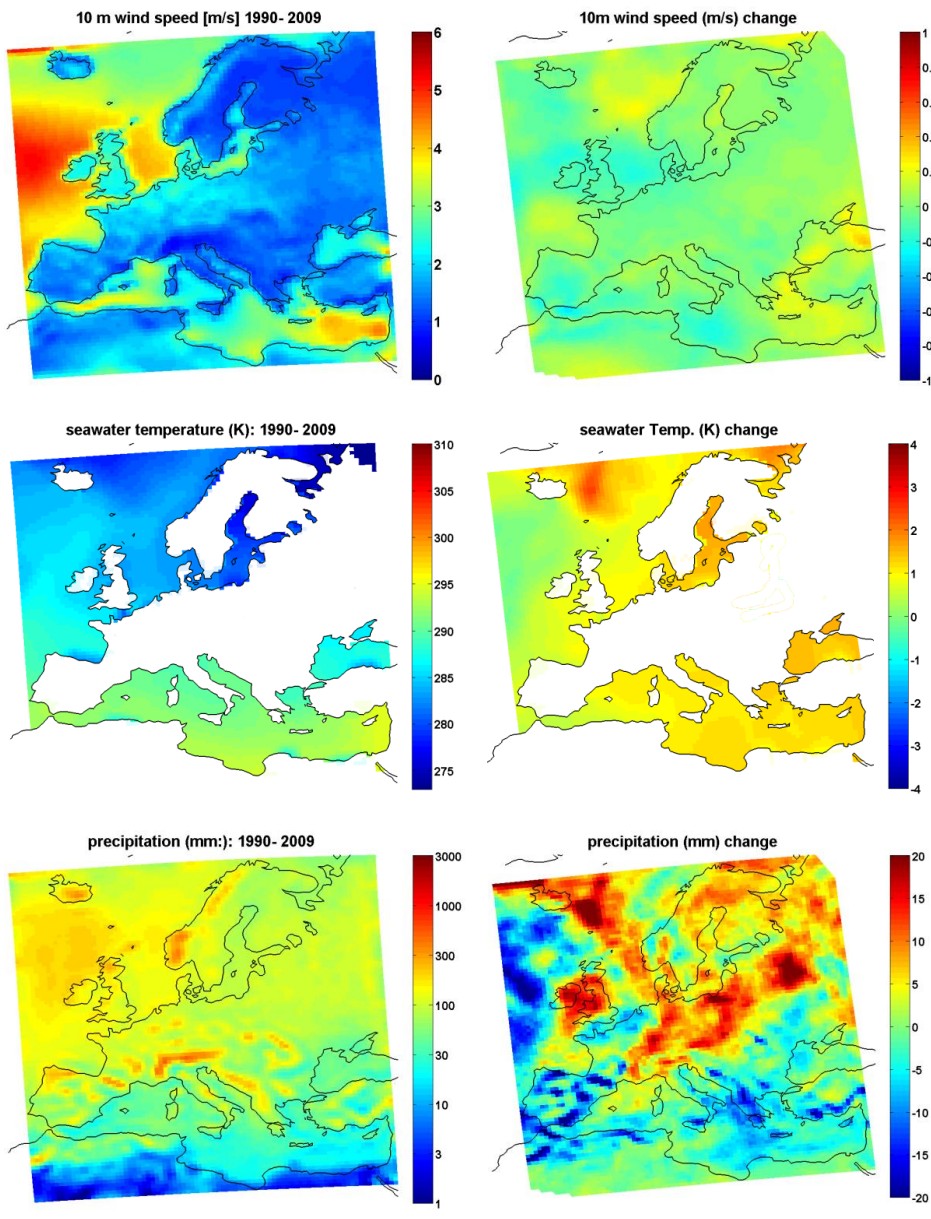


Figure 1. Top: Sea surface temperature (K), middle: wind speed (m s$^{-1}$), bottom: precipitation (mm). Left
panel: mean value for the past period (1990-2009); right panel: absolute difference between the future (2040-
2059) and past periods.





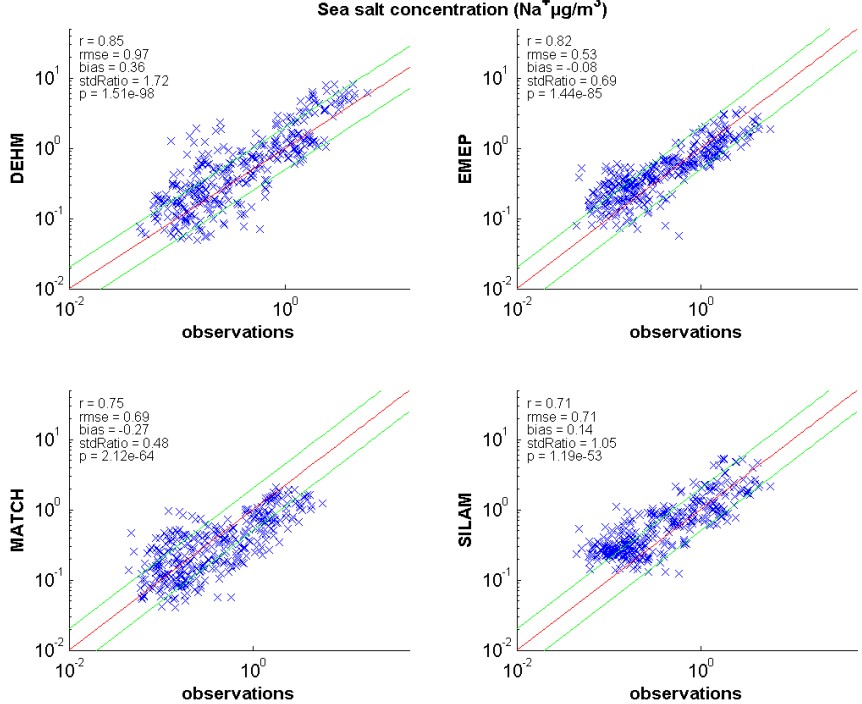

**Figure 2.** Model-measurement comparison for Na$^+$ monthly mean concentration (µg m$^{-3}$) for
29 EMEP measuring sites, between 1990 and 2009. The Person correlation (r), root mean
square error (rmse), bias, standard deviation ration (stdRatio), p-value (p), 1:1 (red solid), 1:2
(green), and 2:1 (green) lines are shown for each CTM.





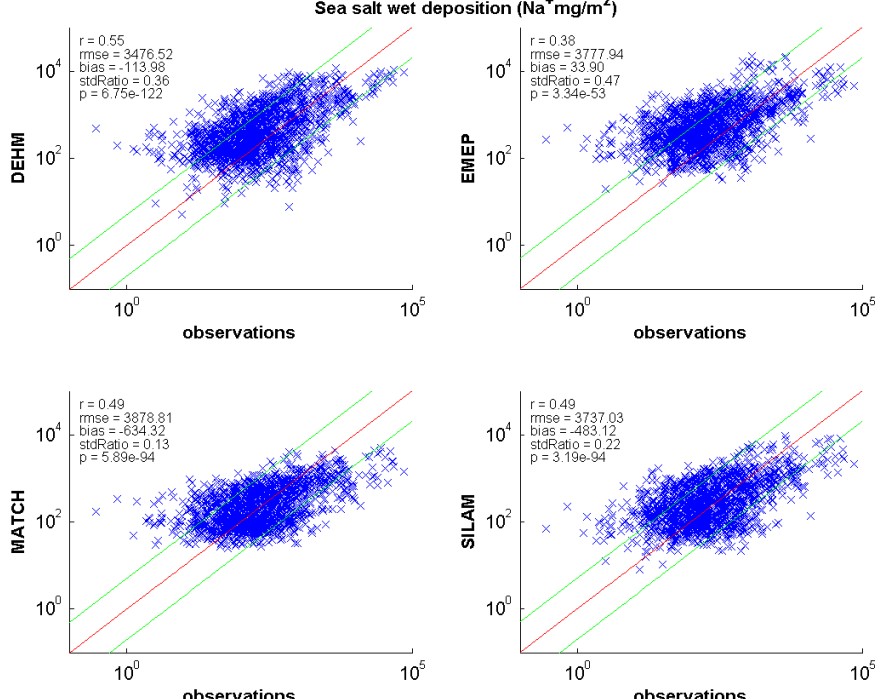

**Figure 3.** Model-measurement comparison for Na[+] monthly wet deposition (µg m[-2]) for 133
EMEP measuring sites, between 1990 and 2009. The Person correlation (r), root mean square
error (rmse), bias, standard deviation ration (stdRatio), p-value (p), 1:1 (red solid), 1:5
(green), and 5:1 (green) lines are shown for each CTM.



**Figure 4.** Annual sea salt emission (mgPM$_{10}$ m$^{-2}$) for DEHM, MATCH and SILAM models.
Left panel mean value for the past period (1990-2009); right panel: absolute difference
between the future (2040-2059) and past periods.



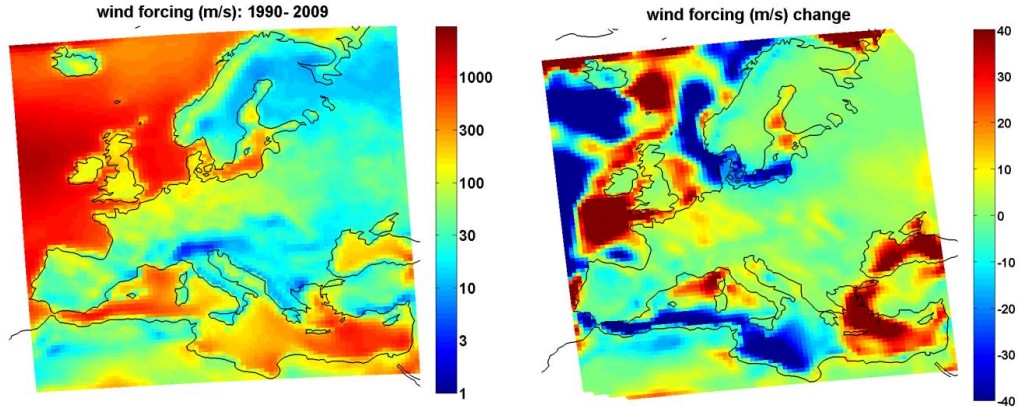

1    **Figure 5.** Wind forcing ($\approx U_{10}^{3.41}$). Left panel: past period (1990-2009); right panel: absolute

2    difference between the future (2040-2059) and past periods.







**Figure 6** Sea salt emission ($mgPM_{10}$ $m^{-2}$) difference between winter (December, January and
February, DJF) and summer (June, July and August, JJA) for DEHM, MATCH and SILAM





1     models. Left panel: past period (1990-2009); right panel: absolute difference between the

2     future (2040-2059) and past periods.







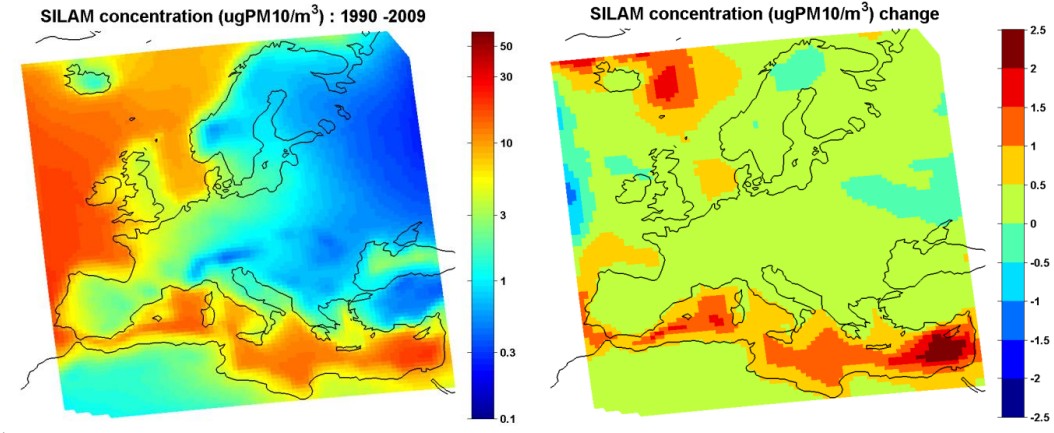

2 **Figure 7**. Sea salt surface concentration ($\mu gPM_{10}$ m$^{-3}$) for DEHM, MATCH, EMEP and

3 SILAM models. Left panel: mean value for the past period (1990-2009); right panel: absolute

4 difference between the future (2040-2059) and past periods.









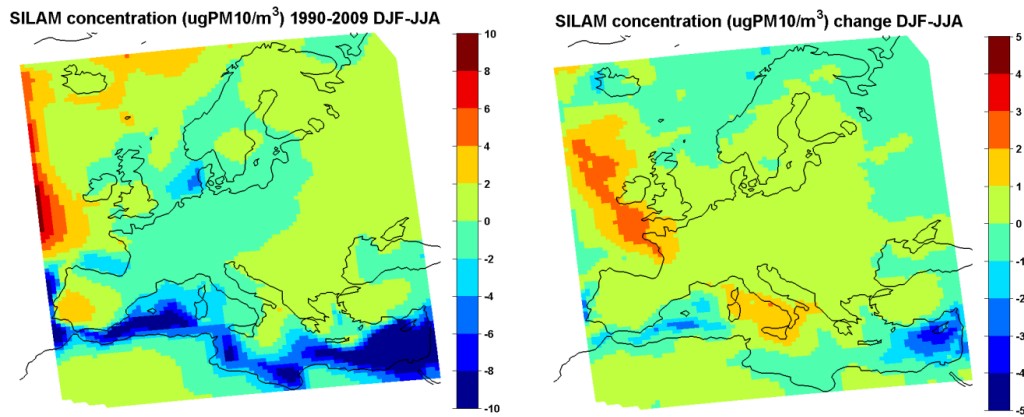

**Figure 8** Sea salt concentration ($\mu$gPM$_{10}$ m$^{-3}$) difference between winter (December, January
and February, DJF) and summer (June, July and August, JJA) for DEHM, MATCH and
SILAM models. Left panel: past period (1990-2009); right panel: absolute difference between
the future (2040-2059) and past periods.











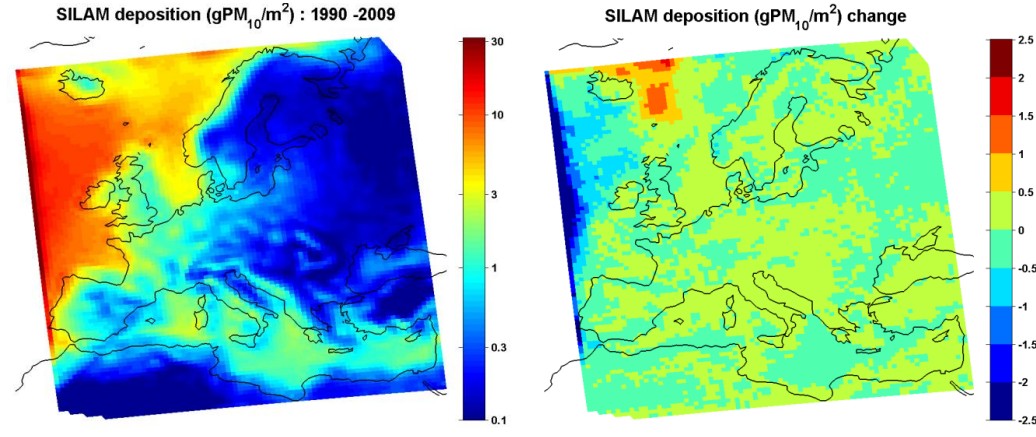

2    **Figure 9** Sea salt deposition (wet+dry) (mgPM$_{10}$ m$^{-2}$) for DEHM, MATCH, EMEP and

3    SILAM models. Left panel: mean value for the past period (1990-2009); right panel: absolute

4    difference between the future (2040-2059) and past periods.







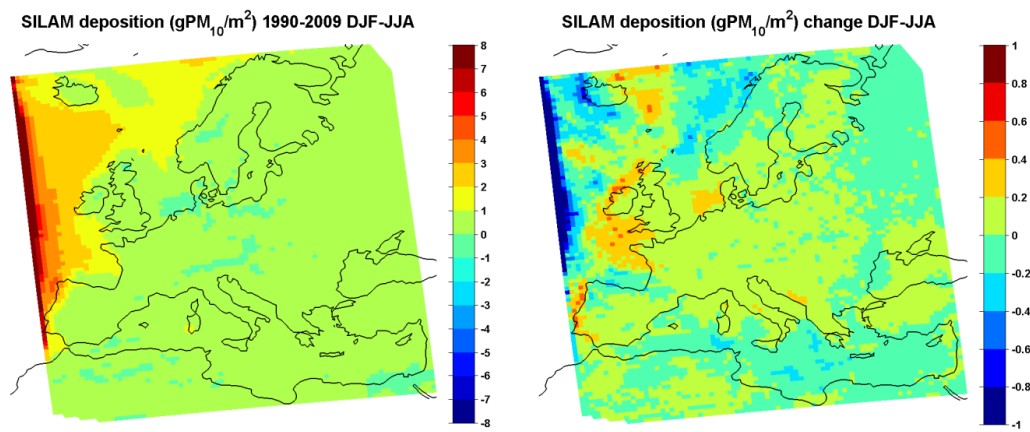

**Figure 10** Sea salt annual deposition (gPM$_{10}$ m$^{-2}$) difference between winter (December,
January and February, DJF) and summer (June, July and August, JJA) for DEHM, MATCH
and SILAM models. Left panel: past period (1990-2009); right panel: absolute difference
between future (2040-2059) and past periods.



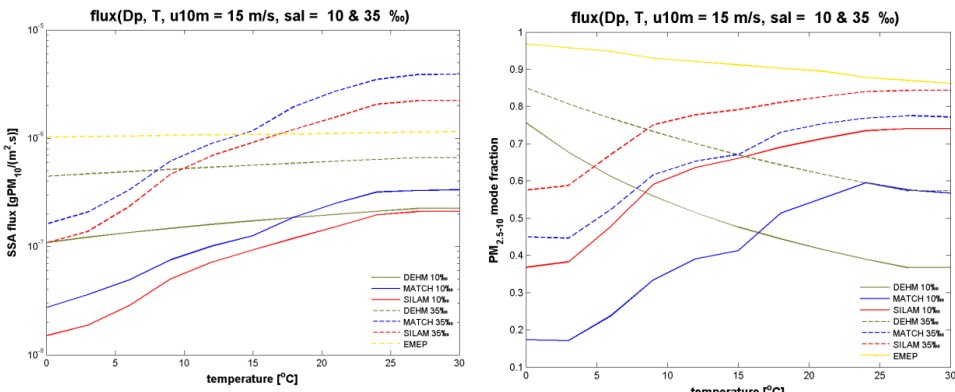

3 **Figure 11.** SSA mass flux [$gPM_{10}$ $m^{-2}$ $s^{-1}$)] box calculations (left) and coarse mode fraction of

4 the mass flux (right): as a function of radius (dry for DEHM and SILAM and RH = 80 % for

5 MATCH) and temperature, for wind speed 15 m $s^{-1}$ and salinities 10 ‰ and 35 ‰.



1

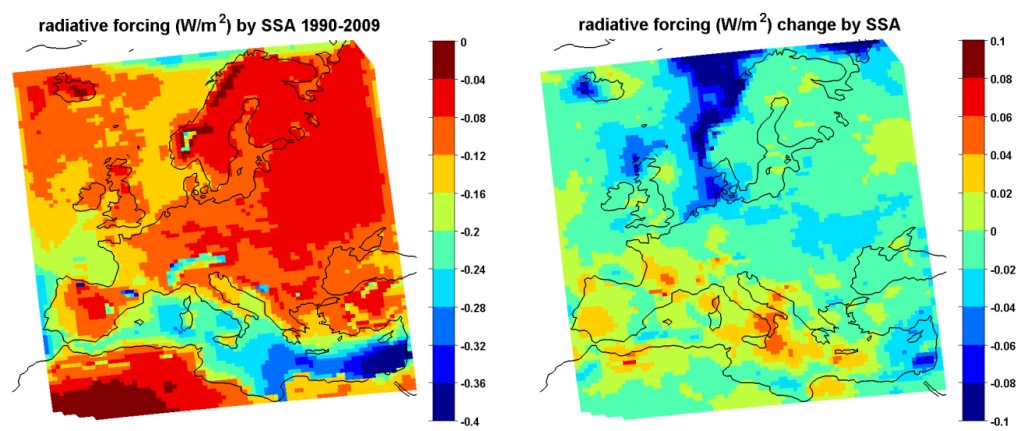

3    Figure 12. Radiative forcing by sea salt (W m$^{-2}$). Left panel: past period (1990-2009); right

4    panel: absolute difference between future (2040-2059) and past periods.



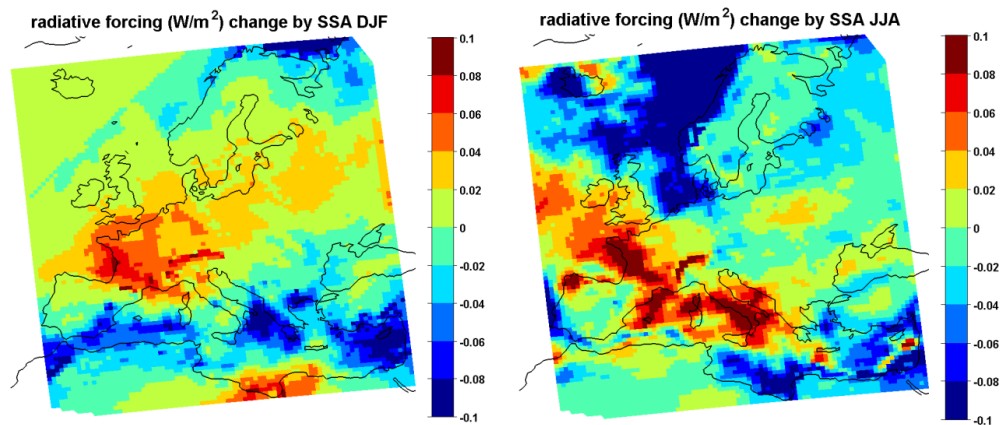

3 Figure 13. Radiative forcing by sea salt (W m$^{-2}$): difference between future (2040-2059) and

4 past periods. Left panel: winter (December, January and February); right panel: summer

5 (June, July and August)

