# Peer review of "Impact of climate change on the production and transport of sea salt aerosol on European seas"

_Atmospheric Chemistry and Physics, 2015_

## Referee Comment (RC1) · Anonymous Referee #1 · 24 Mar 2016

This is an interesting study on the role of an often neglected aerosol such as sea-salt in future climate scenarios. The authors use four different chemistry transport models driven by the same meteorology (climate scenario SRES A1B) and assess the changes in sea-salt emissions, concentrations, depositions and radiative forcing between the periods 1990-2009 and 2040-2059. The manuscript is within the scope of ACP, it is well written and the scientific analysis is thorough and clear. Therefore I recommend publication in ACP after the authors address the following comments:

1. In section 2.3 the authors should make more clear that in DEHM and EMEP the sea salt flux parameterisation assumes a constant 80% RH while MATCH and SILAM use the RH from the climate model. This is stated in Table 1 but should also be emphasized in the text since RH is an important component of the sea salt flux formulation.

2. Including also the basic equations described in sections 2.3.1-2.3.3 (may be in supplement) would help in clarifying the main differences between the models.

3. Lines 169-170: "The fine and coarse fractions in the DEHM model are in the current paper assigned the dry diameters of 1 $\mu$m and 6 $\mu$m". Do you assume a lognormal or some other type of size distribution? Please clarify. The same in lines 192-193, 204-205, 238-239.

4. Please clarify the use of different values for dry/wet sea-salt density (1150 kg/m3, 2200 kg/m3) in the different models.

5. Correct person to pearson Figures 2,3.

6. Line 320 : regional

7. What are the 5 first lines in Table 3?

8. Figure 2 is not needed since the information is already included in Table 3.

9. Line 476: differences

10. Line 555: for all

11. Lines 565-566: "Conversely, warming is predicted where the albedo is high and the AOD is low, e.g. over the mountain tops in Norway and Italy ". I don't see any warming in Fig12a over the mountains. To me it looks like a net cooling of up to -0.28 W/m2 over these areas. In general I am a little confused about the radiative forcing resutls. Less cooling is not warming and to my understanding the effect of sea salt is found to be an overall cooling for both periods. Including also the corresponding radiative forcing plot for the future runs in Figure 12 could clarify this.

12. Line 567: Could you provide an estimation on the uncertainty range for this result?

13. Line 612: Replace "," with ";"

14. Line 615: change on SSA deposition

15. Line 617: questions

16. Lines : 619-621 – not clear please rephrase

17. Line 626: mainly

18. Sea salt particles may serve as CCN for the formation of warm clouds especially in the Mediterranean. Such indirect effects should be also discussed in the analysis.

---

## Short Comment (SC1) · 13 Apr 2016

We would like to thank the reviewer for the positive comments and suggestions. We are thoroughly revising the paper following the recommendations. We will present a response when the comments from other Referee will be available.

---

## Referee Comment (RC2) · Anonymous Referee #2 · 21 Jul 2016

The manuscript presents an interesting study on SSA modelling in present and future scenarios, addressing the role of SSA in the climate system and the current difficulties in modelling it with sufficient accuracy. The manuscript is well written and the results are presented clearly. I recommend publication after the authors address the following (minor) comments:

L120. Correct "predicted an stronger" in "predicted a stronger".

L158. For DEHM, there is a discrepancy in the upper cut of the predicted coarse SSA between the text and Tab. 1 (6 $\mu$m vs 10 $\mu$m), please clarify.

L366. The authors should explain why salinity was kept constant between present and future scenarios. Are salinity changes considered negligible within the considered time horizon? Is it technically impossible to model salinity changes for future scenarios?

Too uncertain?

L413. Remove the comma after "but".

L557. The calculated DRE must depend on assumptions made on the number size distribution of SSA, as radiative properties are driven by particle number and not by mass. This is not very clear in section 2.5, apart a brief note in lines 291-292. How does the SSA number size distribution deployed in libRadtran compares with the different mass distributions predicted by the models and how sensitive is the resulting DRE to changing the SSA number size distribution? The authors should clarify better these issues.

L565. This sentence would be more correct in this way: "Less cooling is predicted where the albedo is higher and SSA is amount is the lowest", as no net warming is observed in Figure12 as an effect of SSA.

L570. Figure 12, not Figure 2.

L570. "The results suggest overall cooling (negative change) in the future": I disagree with this interpretation of Figure 12. It seems clear to me that Europe is neatly divided in two, with cooling in the North and East and warming in the South-West (as it is addressed in the following lines).

L622. "According to this study the upward scattering by SSA, at TOA, can to be up to 0.5 W m-2 over the seawater surfaces in the present period": I would report also the average values over the sea here, as the maximum value is only representative of a very localized situation.

---

## Author Comment (AC1) · 21 Sep 2016

**Response to the comments of the reviewers of the paper "Impact of climate change on the production and transport of sea salt aerosol on European seas" by J. Soares et al.**

We would like to thank the reviewers for their comments and suggestions. We thoroughly revised the paper following these recommendations. Below, we include the responses to each of the issues raised.

**Reviewer #1:**

1. In section 2.3 the authors should make more clear that in DEHM and EMEP the sea salt flux parameterisation assumes a constant 80% RH while MATCH and SILAM use the RH from the climate model. This is stated in Table 1 but should also be emphasized in the text since RH is an important component of the sea salt flux formulation.

✓ This has been revised in the new manuscript

2. Including also the basic equations described in sections 2.3.1-2.3.3 (may be in supplement) could help in clarifying the main differences between the models.

✓ This has been revised in the new manuscript and added as a supplement.

3. Lines 169-170: "The fine and coarse fractions in the DEHM model are in the current paper assigned the dry diameters of 1 µm and 6 µm". Do you assume a lognormal or some other type of size distribution? Please clarify. The same in lines 192-193, 204-205, 238-239.

✓ All the models assume a lognormal distribution. This was included in the general information about the models

4. Please clarify the use of different values for dry/wet sea-salt density (1150 kg/m3, 2200 kg/m3) in the different models.

✓ These values are default values inside of each model. This was not considered has an extremely important factor regarding sea salt, therefore not harmonized between the models

5. Correct person to pearson Figures 2,3.

✓ This has been revised in the new manuscript.

6. Line 320 : regional

✓ This has been revised in the new manuscript.

7. What are the 5 first lines in Table 3?

✓ This has been revised in the new manuscript.

8. Figure 2 is not needed since the information is already included in Table 3.

✓ This has been revised in the new manuscript and the Figure 2 was excluded. Figure 3 was also excluded considering that the same applies for wet deposition.

9. Line 476: differences

✓ This has been revised in the new manuscript.

10. Line 555: for all

✓ This has been revised in the new manuscript.

11. Lines 565-566: "Conversely, warming is predicted where the albedo is high and the AOD is low, e.g. over the mountain tops in Norway and Italy ". I don't see any warming in Fig12a over the mountains. To me it looks like a net cooling of up to -0.28 W/m2 over these areas. In general I am a little confused about the radiative forcing resutls. Less cooling is not warming and to my understanding the effect of sea salt is found to be an overall cooling for both periods. Including also the corresponding radiative forcing plot for the future runs in Figure 12 could clarify this.

✓ This has been revised in the new manuscript. We agree with the reviewer about how "warming" is used in this section, we change the text in order to not to confuse readers. We would prefer to keep Figure 12 as it is for the sake of consistency with the previous pictures.

12. Line 567: Could you provide an estimation on the uncertainty range for this result?

✓ The major source for bias/uncertainty is the input used for the runs (AOD, SSA, etc). The revised manuscript mentions the possible uncertainty related to the estimations for DRE.

13. Line 612: Replace "," with ";"

✓ This has been revised in the new manuscript.

14. Line 615: change on SSA deposition

✓ This has been revised in the new manuscript.

15. Line 617: questions

✓ This has been revised in the new manuscript.

16. Lines : 619-621 – not clear please rephrase

✓ This has been revised in the new manuscript.

17. Line 626: mainly

✓ This has been revised in the new manuscript.

18. Sea salt particles may serve as CCN for the formation of warm clouds especially in the Mediterranean. Such indirect effects should be also discussed in the analysis.

✓ The indirect effect was discussed briefly in the new manuscript.

**Reviewer #2:**

1. L120. Correct "predicted an stronger" in "predicted a stronger".

✓ This has been revised in the new manuscript.

2. L158. For DEHM, there is a discrepancy in the upper cut of the predicted coarse SSA between the text and Tab. 1 (6 m vs 10 m), please clarify.

✓ This has been revised in the new manuscript. It should be in dry diameter in the table, therefore 6μm.

3. L366. The authors should explain why salinity was kept constant between present and future scenarios. Are salinity changes considered negligible within the considered time horizon? Is it technically impossible to model salinity changes for future scenarios? Too uncertain?

✓ Salinity could have been another variable to consider in this study, it was not technically challenging for the dispersion models. However, its change within the discussed time horizon is indeed quite uncertain but presumably small. Keeping salinity constant allows singling out some of the issues with the parameterizations, since temperature is the parameter that changes the most – and for which there is some common ground between the research groups. Also, one of the reasons not to change salinity between periods is that EMEP parameterization does not consider salinity for sea spray production.

4. L413. Remove the comma after "but".

✓ This has been revised in the new manuscript.

5. L557. The calculated DRE must depend on assumptions made on the number size distribution of SSA, as radiative properties are driven by particle number and not by mass. This is not very clear in section 2.5, apart a brief note in lines 291-292. How does the SSA

number size distribution deployed in libRadtran compares with the different mass distributions predicted by the models and how sensitive is the resulting DRE to changing the SSA number size distribution? The authors should clarify better these issues.

✓ DRE computation takes AOD estimated by SILAM, which depends on the assumption on the size distribution described on Table 1.There are several ways to set libRadTran to compute DRE and the way chosen in this study is to describe the aerosol via AOD, therefore the size distribution is taken into consideration and no other information regarding the size distribution is needed. We have considered the effect of size distribution on AOD in connection to fire smoke (Toll et al, AtmEnv, 2015) and, expectedly, found substantial sensitivity. Therefore, the sea salt size spectrum and its dependence on water temperature and salinity are considered quite accurately in SILAM.

6. L565. This sentence would be more correct in this way: "Less cooling is predicted where the albedo is higher and SSA is amount is the lowest", as no net warming is observed in Figure12 as an effect of SSA.

✓ This has been revised in the new manuscript. We agree with the reviewer about the "warming", the terminology has been misused in this section

L570. Figure 12, not Figure 2.

✓ This has been revised in the new manuscript.

L570. "The results suggest overall cooling (negative change) in the future": I disagree with this interpretation of Figure 12. It seems clear to me that Europe is neatly divided in two, with cooling in the North and East and warming in the South-West (as it is addressed in the following lines).

✓ We agree with the reviewer's comment and this has been revised in the new manuscript.

L622. "According to this study the upward scattering by SSA, at TOA, can to be up to 0.5 W m-2 over the seawater surfaces in the present period": I would report also the average values over the sea here, as the maximum value is only representative of a very localized situation.

✓ This has been revised in the new manuscript.